# Non-pharmaceutical interventions for people living with HIV with cognitive impairment: A scoping review

Lucinda Stuart[1]*, Kate Alford[2], Jamie H. Vera[2,3]

1 Brighton and Sussex Medical School, Brighton, United Kingdom, 2 Department of Global Health and Infection, Brighton and Sussex Medical School, Brighton, United Kingdom, 3 University Hospitals Sussex, Brighton, United Kingdom

* L.Stuart1@uni.bsms.ac.uk

**Data Availability Statement:** All relevant data are within the manuscript and its Supporting Information files.

## Abstract

### Background

Cognitive impairment (CI) in HIV is often of multifactorial causation, and remains a prominent issue in the age of effective combination antiretroviral therapy (cART), affecting approximately 14% of people living with HIV. Despite the 2018 BHIVA directive stating the importance of commencing rehabilitation strategies in people living with HIV with CI, no types of cognitive rehabilitations or other non-pharmaceutical interventions are specifically recommended. This scoping review aimed to describe the types of and evidence relating to the non-pharmaceutical interventions which have been examined in people living with HIV with CI.

### Methods

Studies were identified from five electronic databases. Criteria for study inclusion were studies describing a non-pharmaceutical intervention published after 1st January 2000 in English, in a population of adults living with HIV with CI detected at baseline, without significant psychiatric or substance-misuse co-morbidity.

### Results

Fourteen studies met the criteria for inclusion, with the Frascati criteria most commonly used to define CI within participant populations. The median intervention length was 12 weeks (IQR = 6.5). Nine studies investigated interventions with some component of computerised cognitive training (CCT); other interventions included diet, exercise and goal management training. Studies most commonly examined neurocognitive outcomes, but also considered other outcomes including quality of life, depressive symptomatology, intervention acceptability and cART adherence. Eight studies observed improvement in cognition with CCT, with effects often maintained for several weeks post-intervention, however, results were not always statistically significant. Self-reported cognitive improvement and intervention acceptability was high amongst participants completing CCT.

**Funding:** The author(s) received no specific funding for this work.

**Competing interests:** JHV has received honoraria and research grants from and been a consultant or investigator in trials sponsored by Merck, Janssen Cilag, Piramal and Gilead Sciences. He has received sponsorship to attend scientific conferences from Janssen Cilag, Gilead Sciences and AbbVie. This does not alter our adherence to PLOS ONE policies on sharing data and materials. Funding for presentations and investigator sponsor grants but not related to this work.

## Conclusions

There was heterogeneity across studies not only in intervention type, but in diagnostic tools used, the chosen outcome measures and cognitive batteries, making comparison difficult. Findings, however, indicate that CCT interventions may produce benefits in cognition and are acceptable to patients. Further research is required in larger samples, alongside identifying specific intervention components that improve outcomes.

## Introduction

Cognitive impairment (CI) is a prominent issue facing people living with HIV, which occurs more frequently and at younger ages compared to those who are HIV-negative [1]. While severe dementia associated with HIV is uncommon due to the effectiveness of combination antiretroviral therapy (cART), milder forms of cognitive impairment are now more prevalent [2]. Approximately 14% of people living with HIV over 50 years of age have been shown to experience objective CI [3], and older age is identified as a risk factor in both HIV and non-HIV populations [4, 5]. This suggests that the prevalence of CI may rise as the HIV population ages [6].

CI in the context of HIV is commonly referred to as HIV-associated neurocognitive disorder (HAND) [7]. However, over the past decade, there has been growing recognition that CIs in people living with HIV are often multifactorial [8]. For many, these impairments are not solely due to brain injury directly caused by HIV, suggesting that the term "HIV-associated" might not fully encompass the range of factors involved. The International HIV-Cognition Working Group has advocated for distinguishing between brain injuries directly associated with HIV and other causes of brain injury in individuals living with HIV, shifting the focus from strictly neuropsychological testing to considering the broader clinical context [9].

Indeed, while the underlying pathogenic mechanisms involved in the development of CI remain under investigation, aetiology, in the era of effective cART, likely involves a combination of irreversible damage to the central nervous system (CNS) from HIV replication prior to cART initiation, neuroinflammation, ongoing HIV replication in the CNS compartment despite systemic control and the presence of comorbid conditions, lifestyle factors and ageing which have cumulative impacts on cognitive function [8].

Identifying the underlying causes of cognitive decline in people living with HIV is crucial for effective management: it is essential to control HIV infection comprehensively, both in the peripheral and central nervous systems, and to consider the potential for HIV to persist in the cerebrospinal fluid (CSF), termed as CSF escape, which should be investigated and treated if present. Reviewing, modifying, and optimising cART is necessary to address issues such as drug resistance, antiretroviral neurotoxicity, CNS penetration, and treatment adherence. However, when individuals experience cognitive decline despite effective HIV treatment, the causes are often multifactorial. Thus, a holistic approach to management is typically required to address and manage comorbidities and their associated risk factors (i.e. cardiovascular risk factors like smoking, hypertension, diabetes, dyslipidaemia, and obesity to maintain brain health; ageing people living with HIV often face multimorbidity and polypharmacy, increasing the risks of frailty, falls, drug interactions etc.). This approach further emphasizes the importance of lifestyle, social, mental health and educational factors when investigating low cognitive performance [9, 10].

Importantly, the CI seen in the majority of people living with HIV appear to be stable [11, 12], however improvement in cognition has proved difficult to achieve. No specific pharmaceutical treatments are currently available for HIV-associated brain injury and numerous small clinical trials of various compounds have not shown a clear beneficial effect on neurocognitive function [13]. While, optimising the management of comorbidities is an essential requisite for cognitive stability and potential improvement, alternative non-pharmaceutical approaches warrant greater consideration. Indeed, in populations of people without HIV with mild CI, cognitive rehabilitation programmes have been found to improve various cognitive domains including verbal, visual and working memory [14] through posited improved resilience to neurodegeneration [15] and enhanced neuroplasticity [16].

Alternative approaches that target broader indicators of wellbeing, such as quality of life are also important to patients [17]. Research indicates there is a direct link between changes in cognition and in health-related quality of life (HRQL), because CI can be a significant source of anxiety and stress, and perpetuates social isolation [18]. Indeed, people living with HIV and CI report particularly low HRQL: lower than people living with HIV without cognitive issues and lower than people without HIV attending generic memory clinics with mild to moderate dementia [10, 19]. People living with HIV with cognitive issues have reported the key factors driving HRQL to be difficulties performing activities of daily living, high levels of social isolation and low levels of self-efficacy to cope with CI [17].

There is value in examining the literature to identify the types of non-pharmaceutical interventions being used along with the evidence pertaining to each intervention type. Despite some HIV guidelines directives advocating for rehabilitation strategies or services to be offered [20], detailed recommendations do not exist. While an interesting review paper by Chan et al. [21] has described both pharmaceutical and non-pharmaceutical strategies for the management of CI in people living with HIV, to the best of our knowledge no systematically synthesised review exists which specifically examines this topic exists. Mapping or scoping what is known about non-pharmaceutical interventions for people living with HIV with CI will assist in identifying gaps in knowledge and stimulate further consideration and development of these alternate management strategies. Here we describe the extent, range and nature of the body of knowledge on non-pharmaceutical interventions for people living with HIV who experience CI, for the purpose of providing a systematic, synthesised summary of the evidence. With this in mind, this review aims to:

1. Identify the intervention types developed and outcomes assessed by studies examining non-pharmaceutical interventions

2. Provide a picture of the evidence related to specific intervention types and describe the acceptability of interventions to people living with HIV with CI

## Materials and methods

A systematic scoping review methodology was chosen given the paucity of research surrounding non-pharmaceutical interventions specifically for this population. This was based on initial searches of Google Scholar and PubMed, which found that most studies looked at pharmaceutical management options only for people living with HIV. Indeed, studies looking at non-pharmaceutical interventions for CI were often examining these interventions in other patient populations, such as those with dementia or depression. Additionally, we found considerable heterogeneity in the type and design of non-pharmaceutical interventions described for people living with HIV with CI, along with a wide range of reporting outcomes, making a systematic review or meta-analysis methodology inappropriate.

This scoping review utilised the framework outlined by Arksey and O'Malley [22], and the Preferred Reporting Items for Systematic Reviews and Meta-Analyses Extension for Scoping Reviews (PRISMA-ScR) [23] was used to inform the structure and design of the review [S1 Appendix].

## Study selection

**Population.**   Studies were selected for inclusion if participants were adults living with HIV, with measurable CI identified at baseline, defined by standardised diagnostic criteria (e.g. Frascati criteria), neurocognitive batteries, or clinical cut-off scores or rating scales. Studies were excluded if participants had major psychiatric or substance misuse co-morbidities. Studies with paediatric populations were excluded as the mechanisms of impact likely differ between adult and child groups making comparisons between and within intervention types inappropriate.

**Concept and context.**   Studies were included if they examined a non-pharmaceutical intervention within the described population, looking at various quantitative and qualitative outcomes. We excluded studies which did not report outcomes separately for those with measurable CI and those without. Studies looking at the association between CI and certain characteristics or lifestyle behaviours were excluded, as studies were required to include pre-intervention and post-intervention data and not just evidence of correlation. Studies had to be published in English and conference abstracts, study protocols, and review articles were excluded. A grey literature search was not included due to concerns regarding the reliability of descriptions.

## Data sources

Five electronic databases were searched to identify the relevant studies: Medline, Embase, PubMed, PsycINFO and CIANHL. Databases were searched on 25th November 2023, with date limits applied from 1st January 2000 to the current day. This timeframe was chosen given the introduction and access of cART from approximately this timepoint onwards [24]. Screening of studies was first completed via analysis of the title and abstracts, with a second stage of screening completed via full-text analysis. The search terms used for Ovid Medline is described in Fig 1, and this strategy was used with some adaptation for the other four databases.

**Search results.**   Our search strategy yielded 2221 articles from the five databases, with a further 7 articles identified from hand-searching of reference lists. These search results were imported to RefWorks for duplicate screening, where a further 1037 articles were removed, leaving 1184 articles for title and abstract screening. This screening was performed by one author (LS) with additional support and clarification from KA, leading to 52 articles being deemed as suitable for full-text screening. From these articles, 14 were included in the final scoping review, as detailed in Fig 1. Thirty eight articles were excluded; studies were excluded if CI was not observed or objectively assessed in participants at baseline, or where participants had significant co-morbidities or substance misuse disorders. Studies were also excluded if outcomes were not reported separately for those with and without HIV completing an intervention.

## Data extraction

Data was extracted using Microsoft Excel. Information collected included authors, year of publication, setting, recruitment, CI diagnostic method, inclusion and exclusion criteria, sample size, participant characteristics, intervention details, cognitive batteries and outcomes.

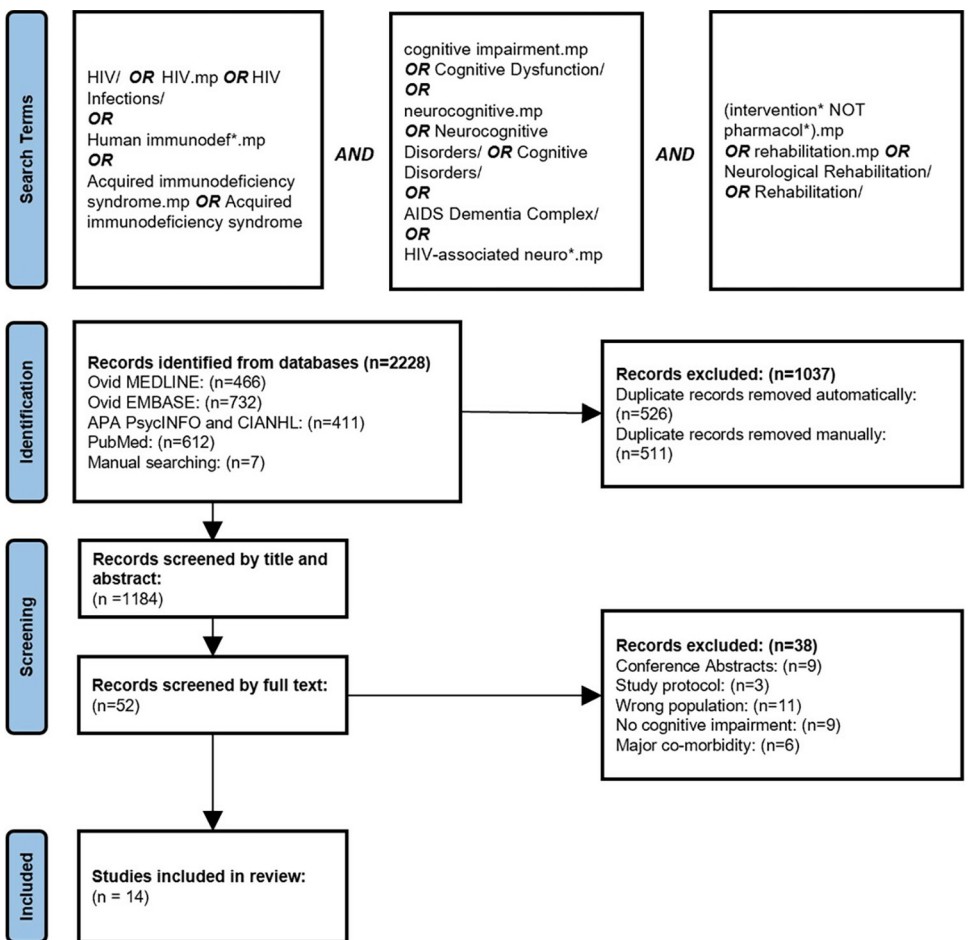

**Fig 1. Search strategy used for Ovid Medline, and PRISMA flowchart describing the identification, screening and inclusion of studies in this scoping review.**

## Results

A total of 14 studies were included in this review, and the key characteristics for each study are reported in detail in Table 1. Ten studies were conducted in the United States [25, 28–30, 32, 34–38], one in Italy [26], one in Iran [27], one in Canada [31], and one in Nigeria [33]. There was a vast heterogeneity of study designs; studies were most commonly single-blinded randomised controlled trials (RCT) [27, 33–35], but also included three pilot RCTs [28, 32, 36], two non-randomised cohort studies [26, 31], one case-comparison study [30], one descriptive cross-sectional study [25], one pre-post experimental design study [37] and one longitudinal RCT [38]. One qualitative study looking at the preliminary experiences of those involved in an ongoing RCT of an exercise intervention was included [29]. Recruitment settings varied from outpatient clinics [26, 32, 37, 38], existing trial cohorts [29, 30, 31, 33], research centres [27], or a combination of settings [25, 28, 35, 36]. Byun et al. [25] looked specifically at participants who had completed the TOPS study, conducted by Vance et al. [37], and Hossain et al. [30] undertook a case-control study on the first three participants to complete each arm of the Think Fast study, performed by Vance et al. [38].

Across all 14 studies, the reported total number of participants randomised was 709, with 589 participants completing the intervention and follow-up periods across studies. The mean

**Table 1. Study characteristics of the 14 studies included in this review.**

| Reference, Year, Country | CI Diagnostic Method | Setting | Sample size and population | Intervention | Intervention Details | Primary Outcomes | Secondary Outcomes | Comparator | Location and Supervision | Adherence |
|---|---|---|---|---|---|---|---|---|---|---|
| Byun et al. 2022, USA [25] | Global Clinical Rating score from neurocognitive battery as determined by experimenters. | Recruited from a HIV clinic, AIDS service organisations and community venues via flyers. | N = 41 (Male = 29, Female = 12) Mean age = 54.6 (6.6) African-American (n = 36,90.0%) | Computerised cognitive training | Individualised targeted computerised cognitive training with Brain HQ. 20 hours of training per week, for 2 domains of objective CI (speed of processing and attention as standard). | Perceived improvement questionnaire | Training satisfaction question | none | n/a | n/a |
| Ceccarelli et al. 2017, Italy [26] | Clinical cut-off score from neurocognitive battery as determined by experimenters. | Recruited from a HIV outpatient clinic in a university hospital. | N = 35 (Male = 33, Female = 2) Mean age = 46.7 (12.4) Caucasian (n = 35, 100.0%) | Probiotic supplement | 6-month course of supplementation with oral probiotics. | Cognitive Battery as determined by experimenters | none | N (I) = 9 N (C) = 26 Control received no treatment. | n/a | Adherence was measured by increase in Bifidobacteria spp. in faecal samples, figures not reported. |
| Etesami et al. 2022, Iran [27]* | Clinical cut-off score in global neurocognitive performance, using the Vienna Test System. | Recruited from a research centre for HIV/AIDS at a university hospital. | N = 60 (Male = 32, Female = 28) Mean age = 37.8 (7.6) | Computerised cognitive training | 8 training modules from CogniPlus completed across twice-weekly 90 minute sessions for 12 weeks. | Vienna Test System | none | N (I) = 30 N (C) = 30 Control received treatment as usual. | Sessions were overseen by two psychologists as clinical trainers. | 81.66% of participants completed the adequate number of intervention sessions. |
| Frain et al. 2017, USA [28]** | Montreal Cognitive Assessment (MoCA) | Recruited from an infectious diseases outpatient clinic and clinical trials unit of a university medical centre. | N = 22 (Male = 19, Female = 3) Mean age = 56.0 (5.2) African-American (n = 13, 59.1%) | Computerised cognitive training | 8-week home-based training programme using BrainHQ software. Training sessions completed 3 times per week for min. 30 minutes. | Montreal Cognitive Assessment (MoCA) | Sleep quality; Depression | N (I) = 10 N (C) = 12 Control received health-related newsletter via email and follow-up phone calls. | Home-based cognitive training. | Self-reported adherence, figures not reported. Retention was 92%. |
| Henry et al. 2016, USA [29] | Frascati Criteria | Recruited from a pre-existing trial cohort. | N = 21 (Male = 18, Female = 3) Mean Age (Int) = 51.8 (2.6) Caucasian (n = 13, 61.9%) | SMS/MMS intervention | 16-week iSTEP personalised and interactive texts to encourage physical activity, with pedometer. | Qualitative interview feedback from questionnaire | Everyday function | N (I) = 11 N (C) = 10 Control received control SMS messages 3 times daily for 16 weeks. | Home-based exercise programme. | In the intervention group, 89% responded to all texts. iSTEP participants reported daily step counts 92% of days. |

*(Continued)*

**Table 1.** (Continued)

| Reference, Year, Country | CI Diagnostic Method | Setting | Sample size and population | Intervention | Intervention Details | Primary Outcomes | Secondary Outcomes | Comparator | Location and Supervision | Adherence |
|---|---|---|---|---|---|---|---|---|---|---|
| Hossain et al. 2017, USA [30] | Frascati Criteria | Recruited from a pre-existing trial cohort. | N = 3 (Male = 2, Female = 1) Mean Age = 54.7 (3.6) African-American (n = 3, 100.0%) | Computerised cognitive training | (1a) 10 hours of speed of processing (SOP) training (1b) 20 hours of speed of processing (SOP) training | Cognitive Battery as defined by experimenters | Visual speed of processing and attention | N (1a) = 1 N (1b) = 1 N (C) = 1 Control received 10 hours of internet training. | Sessions were overseen on-site under the supervision of a trained research assistant. | Each of the three participants were the first in each study arm to complete the allocated intervention. |
| Mayo et al. 2022, Canada [31] | Brief Cognitive Ability Measure (B-CAM) | Recruited from a pre-existing trial cohort. | N = 53 (Male = 49, Female = 4) Mean Age (Int) = 58.1 (7.0) | Goal Management Training | Small group in-person therapy. Participants met in groups of 5–8 for 120 minute training sessions for 9 weeks. | Brief Cognitive Ability Measure (B-CAM) | Self-reported cognitive ability | N (I) = 30 N (C) = 23 Control was an untreated group for another pilot RCT. | Supervised group sessions with 5-8 participants. | 70% of participants were classified as high adherence for analysis. |
| Morrison et al. 2020, USA [32] | Modified Telephone Interview for Mild Cognitive Impairment (TICS-M) | Recruited from university's HIV clinic. | N = 14 (Male = 6, Female = 8) Mean Age (Int) = 56.7 (5.7) | Ketogenic diet | Participants selected meals from an 8-day ketogenic diet menu designed by a dietician, for 12 weeks. | Cognitive Battery as defined by experimenters | Intervention fidelity; Anthropometric measures; Serum biomarkers; Metabolic indicators | N (I) = 7 N (C) = 7 Control received treatment as usual (patient-choice diet). | n/a | 35.7% of participants assessed capillary ketones as instructed. |
| Nweke et al. 2022, Nigeria [33] | Frascati Criteria | Recruited from a pre-existing study population. | N = 73 Age, gender and ethnicity data not available. | Aerobic exercise structured programme | Aerobic exercise sessions, completed for 20–60 minutes 3 times per week for 12 weeks. Exercise was moderate intensity using a cycle ergometer. | Quality of life | ART adherence; Depression; Oxygen Saturation; Heart rate; Blood pressure (SBP and DBP) | N (I) = 38 N (C) = 35 Control received physical activity advice and education. | Sessions were conducted on-site and conducted over several weeks, and were not directly supervised. | Completion was 91.78%. |
| Ownby et al. 2016, USA [34] ** | Frascati Criteria AND Clinical cut-off score from neurocognitive battery as determined by experimenters. | Recruited from a pre-existing study population and local organisations. | N = 11 (Male = 9, Female = 2) Mean Age = 51.5 (4.7) African-American (n = 2, 18.2%) | Computerised cognitive training AND transcranial direct stimulation (tDCS) | Computerised cognitive training (with GT Racing 2 computer game) with active tDCS (20 minutes active stimulation at 1.5 mA). Participants took part in 6 training sessions over 2 weeks. | Cognitive Battery as defined by experimenters | Self-reported function; Self-reported depressive symptoms | N (I) = 6 N (C) = 5 Control received computerised cognitive training (with sham tDCS (30 seconds active stimulation at 1.5 mA). Participants took part in 6 training sessions over 2 weeks. | Sessions were overseen by an investigator who sat at another desk behind the participant. | 78.57% of participants completed all study procedures. |

*(Continued)*

**Table 1.** (Continued)

| Reference, Year, Country | CI Diagnostic Method | Setting | Sample size and population | Intervention | Intervention Details | Primary Outcomes | Secondary Outcomes | Comparator | Location and Supervision | Adherence |
|---|---|---|---|---|---|---|---|---|---|---|
| Ownby et al. 2021, USA [35] * | Clinical cut-off score from neurocognitive battery as determined by experimenters. | Recruited from a pre-existing study population and local service providers for people living with HIV. | N = 46 (Male = 37, Female = 9) Mean Age (Ia) = 57.1 (5.0) African-American (n = 29, 63.0%) | Computerised cognitive training AND transcranial direct stimulation (tDCS) | (Ia) Computerised cognitive training (with GT Racing 2 computer game) with active tDCS (20 minutes active stimulation at 1.5 mA). (Ib) Computerised cognitive training (with GT Racing 2 computer game) with sham tDCS (30 seconds active stimulation at 1.5 mA). | Cognitive Battery as defined by experimenters | Acceptability of intervention; Participant reaction time; Everyday functional performance | N (Ia) = 16 N (Ib) = 15 N (C) = 15 Control received educational videos with sham tDCS. | Sessions were overseen by an unblinded principal investigator. | 95.65% of participants completed the study. |
| Towe et al. 2017, USA [36] | Working Memory Test Battery | Recruited from community-based organisations and infectious diseases clinics. | N = 21 (Male = 16, Female = 5) Mean Age = 47.9 (11.2) African-American (n = 18, 86%) | Computerised cognitive training | 5 memory training tasks from PSSCogRehab, completed in 12 training sessions over 10 weeks. | Working Memory Test Battery | Speed of processing; Memory; Executive function; Verbal fluency; Motor skills | N (I) = 11 N (C) = 10 Control received attention-matched control training. | Sessions were conducted on-site and conducted over several weeks, and were not directly supervised. | 100% of the intervention group completed all sessions. |
| Vance et al. 2021, USA [37] | Global Clinical Rating score from cognitive battery | Recruited from a university HIV/AIDS clinic. | N = 88 (Male = 60, Female = 28) Mean Age = 54.2 (7.0) African-American (n = 75, 85.2%) | Computerised cognitive training | Individualised targeted computerised cognitive training with Brain HQ. 20 hours of training per week, for 2 domains of objective CI.Speed of processing and attention tested as standard, algorithm used to determine two domains most likely to reverse a HAND diagnosis if these deficits not present. | Cognitive Battery as defined by experimenters | none | N (I) = 48 N (C) = 40 Control received no contact or treatment. | Sessions were conducted on-site and supervised by a cognitive trainer. | 64% of the intervention group completed the training. |

*(Continued)*

**Table 1.** (Continued)

| Reference, Year, Country | CI Diagnostic Method | Setting | Sample size and population | Intervention | Intervention Details | Primary Outcomes | Secondary Outcomes | Comparator | Location and Supervision | Adherence |
|---|---|---|---|---|---|---|---|---|---|---|
| Vance et al. 2023, USA [38] ** | Global Clinical Rating score from neurocognitive battery as determined by experimenters. | Recruited from a university HIV/AIDS clinic. | N = 114 (Male = 71, Female = 43) Mean Age = 51.5 (6.3) Black, Indigenous and People of Colour = (n = 97, 85.1%) | Computerised cognitive training | (1a) Speed of processing training with Brain HQ for 10 hours. (1b) Speed of processing training with Brain HQ for 20 hours. Training was completed over 10–20 weeks. | Cognitive Battery as defined by experimenters | HAND status | N (1a) = 70 N (1b) = 73 N (C) = 73 Control received Internet Navigation Control Training for 10 hours. | Sessions were overseen on-site under the supervision of a trained research assistant. | The average number of SOP hours engaged in training was 8.88 (SD = 2.96) hours and 18.25 (SD = 5.22) hours for the 10-h group and 20-h group. |

* ~ intention-to-treat analysis completed participant baseline characteristics and per-protocol analysis completed for outcomes.

** ~ per-protocol analysis completed for both participant baseline characteristics and outcomes.

age of participants reported across studies was 52.2 years (SD = 5.53). One study [33] did not report demographic information, and four studies [29, 31, 32, 35] reported the mean age per subgroup and not as a total cohort; for these studies, the mean age of the primary intervention subgroup was used to calculate the mean age of all participants in this review, and are included in Table 1. The majority of participants across studies were male, and African-American or Caucasian ethnicities were the most commonly represented across all studies. The median intervention length was 12 weeks (IQR = 6.5), and this ranged from 2 to 24 weeks across studies. The exact details of the intervention duration, regularity and intensity are detailed in Table 1, and full demographic data is available in S1 Table. The median follow-up timepoint was 12 weeks (IQR = 14) following intervention initiation, but follow-up timepoints varied from 3 weeks to 2 years from baseline.

The diagnostic method used to identify CI at baseline, and determine the eligibility of participants, varied across all studies. Four studies used the Frascati criteria to detect CI; other studies used global clinical rating scores or clinical cut-off scores from each neurocognitive battery to detect impairment in cognitive function. Other tools used are detailed in Table 2. Thirteen studies included a control group, and the details of comparators are also included within Table 1.

The majority of studies (n = 11) examined cognitive function as the primary outcome. A detailed description of the neurocognitive batteries used across studies, including the assessment tools used to assess different cognitive domains, are outlined in Table 2. Other outcomes examined by studies included QoL, depressive symptomatology, intervention acceptability and cART adherence; these are detailed in Table 3.

## Computerised Cognitive Training (CCT)

CCT is a cognitive rehabilitation technique that uses computerised training programmes and exercises to target and improve specific cognitive domains, with the hope of improving cognitive function in these areas [39]. CCT either focuses on one specific cognitive domain via targeted training or aims to improve several domains with various exercises. Nine studies in this review looked at interventions with some component of CCT.

Etesami et al. [27] conducted a single-blinded RCT with the CogniPlus® training package, involving weekly 90-minute sessions over 12 weeks. The study found improvements in six cognitive domains and overall neurocognitive performance, sustained for three months post-intervention. No cognitive changes were observed in the control group, indicating the potential effectiveness of CCT in improving cognitive function. Frain et al. [28] examined an 8-week home-based CCT program with sessions three times a week. Participants in the intervention group showed statistically significant cognitive improvements immediately post-intervention and up to 16 weeks later, particularly in executive function. However, these gains were not sustained over time, as the Montreal Cognitive Assessment scores showed no significant difference between the intervention and control groups at 16 weeks.

The Training on Purpose Study (TOPS) examined 20 hours of individualized CCT over 12 weeks, targeting specific cognitive deficits identified at baseline. Where participants had deficits in speed of processing or attention, these were targeted as standard; if these deficits were not present, an algorithm selected two domains to train that would most likely reverse a HAND diagnosis for that participant. While the study did not reverse a HAND diagnosis for most participants, it did improve global clinical ratings [37]. Participants who received speed of processing (SOP) training reported improvements not only in SOP but also in other cognitive domains, though these changes were not statistically significant. Byun et al. [25] evaluated self-reported perceived improvement experienced by participants in the TOPS study and

**Table 2. Neurocognitive batteries used across studies in this scoping review, where relevant, with a description of the assessment tools used to assess specific cognitive domains.**

| Study | Assessment Tools | Cognitive Domain |
|---|---|---|
| Ceccarelli et al. 2017, Italy [26] | Rey-Osterrieth Complex Figure Test (ROCF) | Recognition and recall skills for non-verbal contents |
| | Rey Auditory Verbal Learning Test (RAVLT) | Short-term auditory-verbal memory, and rate of learning and retention of information |
| | Test of Weights and Measures Estimation (STEP) | Abstraction skills |
| | Visual Search Test (Attention Matrices Test) | Attention skills |
| | Verbal Fluency Test (FAB) | Executive functions and ability to switch between different tasks |
| | Test of Phonological and Semantic Verbal Fluency (PVF and SVF) | Phonological and semantic supplies and abilities to access them |
| | Raven's Standard Progressive Matrices (SPM) | Abstract reasoning and problem-solving capabilities |
| | Wechsler Adult Intelligence Scale, 4th edition (WAIS-IV), Digit Span subtest | Short-term memory and executive functions/attention |
| | Corsi Block Tapping Test (CBTT) | Short-term spatial memory and executive functions |
| | Trail Making Test A (TMT A) | Visual-spatial attention and motor skills, speed of information processing/psychomotor speed |
| | Trail Making Test B (TMT B) | Executive function |
| | Aachener Aphasia Test (AAT) | Prescence of aphasia |
| Etesami et al. 2022, Iran [27] | Recurring Figures Test | Non-verbal learning |
| | Tower of London (Freiburg Version) | Planning ability |
| | Visual Memory Test | Visual memory |
| | Divided Attention Test | Divided attention |
| | Spatial Attention (Vigilance) Test | Spatial attention |
| | Selective Attention Test | Selective attention |
| Frain et al. 2017, USA [28] | Montreal Cognitive Assessment (MoCA) | Attention |
| | | Concentration |
| | | Executive function |
| | | Memory |
| | | Language |
| | | Conceptual thinking |
| Hossain et al. 2017, USA [30] | Controlled Oral Word Association Test | Verbal Fluency |
| | Animal and Action | |
| | Wisconsin Card Sorting Test | Executive functions |
| | Trail Making Test B | |
| | Trail Making Test A | Speed of information processing |
| | Digit Symbol Task | |
| | Symbol Search | |
| | Hopkins Verbal Learning Test | Learning and delayed recall |
| | Brief Visuospatial Memory Test | |
| | Letter Number Sequencing | Attention and working memory |
| | Paced Auditory Serial Addition Test | |
| | Grooved Pegboard Test dominant and non-dominant hands | Motor skills |

*(Continued)*

**Table 2.** (Continued)

| Study | Assessment Tools | Cognitive Domain |
|---|---|---|
| Mayo et al. 2022 [31] | Brief Cognitive Ability Measure (B-CAM) | Processing speed |
| | | Attention |
| | | Memory |
| | | Executive function |
| | Tower of London Test | Working memory and attention as applied to multi-step planning and problem-solving |
| Morrison et al. 2020 [32] | Trail Making Test A | Psychomotor speed |
| | Digit Symbol Substitution Test (DSST) | Speed of information processing |
| | Trail Making Test B | Executive function |
| | Stroop Colour-Word Naming Task | Executive function and inhibition |
| | Hopkins Verbal Learning Test (HVLT-R) | Verbal memory |
| Ownby et al. 2016 [34] | Wechsler Adult Intelligence Scale, 4th edition (WAIS-IV), Digit Span subtest | Attention and Working Memory |
| | Trail Making Test A | Executive function |
| | Trail Making Test B | |
| | Hopkins Verbal Learning Test (HVLT-R) | Verbal learning and memory |
| | Grooved Pegboard Test dominant and non-dominant hands | Psychomotor speed |
| Ownby et al. 2021 [35] | Wechsler Adult Intelligence Scale, 4th edition (WAIS-IV), Digit Span subtest | Attention |
| | Wechsler Memory Scale, 4th edition, Symbol Span subtest | |
| | Wechsler Adult Intelligence Scale, 4th edition (WAIS-IV), Coding Subtest | Psychomotor speed |
| | Trail Making Test A | |
| | Grooved Pegboard Test dominant and non-dominant hands | |
| | Trail Making Test B | Executive function |
| | Verbal design and fluency from Delis-Kaplan Executive Function System (D-KEFS) | |
| | Stroop Colour Word Test | |
| | Iowa Gambling Task | |
| | Hopkins Verbal Learning Test (HVLT-R) | Learning and memory |
| | Brief Visuospatial Memory Test | |
| Towe et al. 2017 [36] | Paced Auditory Serial Addition Task-50 | Working Memory |
| | NAB Digits Forward / Digits Backward Test | |
| | Trail Making Test A | Speed of information processing |
| | Hopkins Verbal Learning Test (HVLT-R) | Learning (immediate recall) and memory (delayed recall) |
| | Stroop Colour Word Test | Executive function |
| | Trail Making Test B | |
| | FAS Letter Fluency | Verbal fluency |
| | Grooved Pegboard Test dominant and non-dominant hands | Motor skills |

(*Continued*)

**Table 2.** (Continued)

| Study | Assessment Tools | Cognitive Domain |
|---|---|---|
| Vance et al. 2021 [37] | Stroop Colour Naming Test | Speed of information processing |
| | Trail Making Test A | |
| | Paced Auditory Serial Addition Test | Attention |
| | Stroop Interference | Executive function |
| | Trail Making Test B | |
| | Benton Visual Retention Test—Revised | Spatial learning and memory |
| | Benton Visual Retention Test Delayed—Revised | Delayed spatial learning and memory |
| | Wechsler Adult Intelligence Scale, 4th edition (WAIS-IV), Block Design subset | Spatial visualisation |
| | Hopkins Verbal Learning Test (HVLT-R) | Verbal learning and memory |
| | Hopkins Verbal Learning Test Delayed (HVLT-R) | Delayed verbal learning and memory |
| Vance et al. 2023 [38] | Controlled Oral Word Association Test | Verbal Fluency |
| | Animal and Action | |
| | Wisconsin Card Sorting Test | Executive functions |
| | Trail Making Test B | |
| | Trail Making Test A | Speed of information processing |
| | Digit Symbol Task | |
| | Symbol Search | |
| | Hopkins Verbal Learning Test | Learning and delayed recall |
| | Brief Visuospatial Memory Test | |
| | Letter Number Sequencing | Attention and working memory |
| | Paced Auditory Serial Addition Test | |
| | Grooved Pegboard Test dominant and non-dominant hands | Motor skills |

reported 76.9% of participants felt their ability to complete their everyday activities had improved moderately or more, and 84.6% felt that the training had improved their SOP. Training satisfaction was high at 94.9%, indicating that CCT interventions were highly acceptable to patients

Speed of processing (SOP) training was further examined in a 2-year, longitudinal trial (the Think Fast Study) [38]. Participants (N = 216) were randomised to receive either 10 or 20 hours of speed of processing training, or active computer-based control, with training completed over 10 to 20 weeks. In contrast to the findings of TOPS, this study found lower global clinical rating scale scores (indicating better cognitive function) in the control group both immediately and one year post-intervention when compared with those in the intervention arms. Furthermore, reversal of HAND status post-intervention was higher in the control group. These findings contradicted initial hypotheses made by the authors; given that this study was longitudinal and has the largest sample size of studies in this review, this may be important for considering the applicability of CCT training in this cohort. Hossain et al. [30] undertook a case-control study of the first three participants in each arm of the Think Fast Study, and found that only the participant receiving 20 hours of speed of processing training did not meet HAND criteria following the intervention period. Given the preliminary nature of this case-control study, these results should be interpreted with caution.

A pilot RCT conducted by Towe et al. [36] delivered 12 CCT training sessions to investigate the effect upon working memory function. An improvement in mean working memory scores

**Table 3. Further outcomes and assessment tools detailed in studies in this review.**

| Study | Outcome | Assessment Tool |
|---|---|---|
| Byun et al. 2022, USA [25] | Self-reported perceived improvement in cognition and everyday function | Perceived improvement questionnaire |
| | Satisfaction | Training satisfaction question |
| Frain et al. 2017, USA [28] | Sleep quality | Pittsburgh Sleep Quality Index (PSQI) |
| | Self-reported depressive symptoms | Center for Epidemiological Studies Depression Scale (CES-D) |
| Henry et al. 2016 [29] | Everyday function | Patient's Assessment of Own Functioning (PAOF) |
| | | Instrumental Activities of Daily Living (IADL) |
| Hossain et al. 2017 [30] | Visual speed of processing and attention | Useful Field of View test |
| Mayo et al. 2022 [31] | Self-reported cognitive ability | Communicating Cognitive Concerns (C3Q) |
| Nweke et al. 2022 [33] | Quality of life | World Health Quality of Life Questionnaire (WHOQoL)-BREF |
| | Depression | Beck Depression Inventory (BDI) |
| | ART adherence | 3-day self-reported history |
| Ownby et al. 2016 [34] | Self-reported cognitive difficulty | Patient's Assessment of Own Functioning (PAOF) |
| | Self-reported depressive symptoms | Center for Epidemiological Studies Depression Scale (CES-D) |
| Ownby et al. 2021 [35] | Functional status | Medication Management Test–Revised |
| | | University of California San Diego performance-based skills assessment |
| | Reaction time | California Computerised Assessment Package |
| | Acceptability of intervention | Questionnaire based on Technology Acceptance Model |

from baseline was observed in active CCT group, but improvement was limited to this cognitive function, demonstrates that CCT training may only yield improvement in the specific domains targeted by training programmes. Participants reported high satisfaction with the intervention, and there was 100% completion rate.

## CCT & transcranial direct current stimulation

In addition to the studies described above, two studies in this review looked specifically at the use of CCT in combination with transcranial direct current stimulation (tDCS). This technique involves delivering a small amount of current to certain brain regions via scalp electrodes, with the aim of influencing neuronal activity and subsequent brain functioning [40]. An initial pilot study [34] looked at the effects of completing CCT alongside 20 minutes of active tDCS with 1.5 milliamps (mA) of current across 6 training sessions, in comparison with placebo tDCS. Findings indicated that active tDCS and CCT improved learning, memory and motor speed in comparison to the control group receiving placebo tDCS and CCT, which indicates a possible role for tDCS to boost the possible positive effects of CCT. Furthermore, the measure of self-reported cognitive difficulty decreased in the active tDCS group, indicating that participants felt the intervention had been beneficial to their cognitive function. In addition, a second, larger study [35] compared active tDCS and CCT with placebo tDCS and CCT, with a further control receiving just sham tDCS alone. Whilst reaction time was not improved

with active tDCS and CCT, there was a significant difference (p = 0.02) between Wechsler Adult Intelligence Scale Fourth Edition (WAIS IV) Coding subtest scores between the active tDCS and control group post-intervention, indicating a possible improvement in psychomotor speed. Beyond this, there was limited evidence to indicate a broader positive impact of the intervention on cognitive function, however, 93% of participants were satisfied with the intervention, and rated the ease of use and enjoyment highly. Furthermore, improvements in psychomotor speed were sustained at 1-month follow-up, indicating a possible longevity of intervention benefit.

## Diet & supplementation

Two studies in this review looked at diet and supplementation as a non-pharmaceutical intervention. One study [32] looked at the feasibility of a ketogenic diet intervention in a small cohort of people living with HIV with CI. Participants selected meals from an 8-day ketogenic diet menu designed by a dietician, for 12 weeks, and were compared with a patient-choice diet control group. Immediately post-intervention, the ketogenic diet group had significantly improved scores on Trails Making Test A (psychomotor speed) ($p = 0.035$) and B (executive function) ($p = 0.039$) compared with controls, but these effects did not continue after a 6-week washout period. Furthermore, the adherence to the intervention was unclear, as only 5 participants assessed capillary ketones consistently to monitor adherence across the study.

Ceccarelli et al. [26] looked at the effect of 6-months probiotic supplementation in 9 patients, compared with controls. They found an improvement in scores across all tests in the cognitive battery in the probiotic group, however this improvement was not always statistically significant when compared with controls. Furthermore, it should be noted that controls were not well-matched to the intervention group, as participants were allocated to the intervention arm if they had higher levels of neuroinflammation as assessed by CSF neopterin concentration.

## Exercise

Two studies in this review looked at exercise-based interventions. Nweke et al. [33] conducted a RCT investigating the effects of an aerobic exercise intervention in people living with HIV with CI; the intervention comprised of 3-times weekly sessions using a cycle ergometer over 12 weeks. Participants who completed the exercise program had higher scores for physical ($p<0.001$) and overall ($p = 0.001$) QoL compared with controls. Participants in the intervention group also reported significantly reduced depressive symptomatology, and this effect persisted for 3 months after completion of the program. For those completing the exercise program, ART adherence peaked immediately following completion of the program.

The iSTEP intervention comprised of 16-week SMS/MMS intervention where participants received personalised and interactive texts encouraging physical activity. Henry et al. [29] gathered data, including qualitative feedback, on the intervention in relation to a control group receiving control SMS messages. In the iSTEP group, there was high engagement, with 92% reporting daily step counts, and 90% of participants stated they were "satisfied" or "very satisfied" with the intervention. All intervention group participants believed the intervention had improved their physical activity levels, and agreed they would recommend the intervention to other people living with HIV. Interestingly, two-thirds of iSTEP participants self-reported an improvement in cognitive function. Participants also indicated that using a pedometer, milestone messages and using step count goals were helpful for increasing physical activity levels.

### Goal management training

One study [31] looked at the effects of Goal Management Training (GMT), a specific form of cognitive rehabilitation that seeks to improve executive functioning via 20 hours training in mindfulness, self-management and other areas [41]. The intervention comprised of small group therapy sessions held across 9 weeks, compared against an untreated control group from another pilot RCT. Results showed no significant change in cognitive measures from baseline to post-intervention in those taking part in GMT, regardless of adherence to the programme, with only 1 participant from the intervention group surpassing the threshold for the reliable positive change with the Reliable Change Index. Additionally, 30% of participants were considered to have low adherence to the intervention, with three participants attending 4 or fewer out of the 9 organised sessions; however, in those with high adherence, there was an increase in self-reported cognitive ability following GMT training.

## Discussion

As a greater proportion of people living with HIV reach older age, addressing CI in this cohort will become increasingly important. Beyond adjusting medication regimes and managing the impact of comorbidities to reduce adverse cognitive effects, there are no pharmaceutical interventions that directly improve cognition or ameliorate QoL for people living with HIV who experience CI. This indicates a role for non-pharmaceutical interventions, which aside from potentially improving neuropsychological outcomes may be effective at improving self-reported functional abilities and QoL. The objective of this scoping review was to describe the non-pharmaceutical interventions for CI in people living with HIV, and describe the current evidence relating to these varied strategies.

CCT appears to have high levels of satisfaction and acceptability in this cohort, although the evidence pertaining to neuropsychological improvement is somewhat mixed in people living with HIV with CI. Interestingly, a recent meta-analysis of CCT in those with mild cognitive impairment (without HIV) found improvements in verbal, visual and working memory [14] with similar findings reported in those with major depression [42], and in dementia [43]. CCT has many practical benefits including the ability for standardisation, precise monitoring of adherence and individual personalisation [44], possibly indicating ease of widespread implantation. Whilst CCT is often carried out with supervision from a trained professional, unsupervised CCT interventions could be administered in home settings to further facilitate access to treatment and reduce costs and resource usage [45]. Further, large scale RCTs are required to clarify the efficacy of CCT in those with HIV and CI and to ascertain whether findings documented in non-HIV populations can be replicated. Additionally, studies should consider incorporating a tDCS arm into research designs given tentative indications that it may boost the effects of CCT.

The review highlights the importance of identifying specific cognitive training targets to enhance cognitive benefits. The ACTIVE study showed that speed of processing (SOP) training in older adults not only improved cognitive function but also greatly reduced the risk of depressive symptoms for several years post intervention [46]. This is relevant for people living with HIV who often exhibit SOP deficits [38] and are far more likely to experience depression than the general population [47]. Moreover, the deterioration of SOP ability with ageing places greater burden upon other cognitive functions such as executive functioning [48]; as a result, any improvement in SOP derived from CCT interventions may span across multiple cognitive domains. While several studies focused on SOP as a therapeutic target, further research is needed to optimize these interventions for people living with HIV.

Exercise may be a key target area of non-pharmaceutical interventions for CI, given that higher levels of physical activity have been associated with less CI in people living with HIV [49–52], and that this review demonstrated beneficial impacts of exercise on QoL and medication adherence. Depression is known to be up to three times higher in people living with HIV compared with the general population [47], which is significant given the reduced depressive symptomatology observed following an exercise intervention in this review [33]. This highlights a role for exercise, as a tool to not only improve cognitive function, but also alleviate depressive symptoms and improve cardiovascular health. The positive effects of exercise interventions on cognitive function has already been demonstrated for various subtypes of dementia [53], however acceptability and long-term adherence plague these intervention types. Further studies which incorporate implementation methodologies are required to investigate the effects of exercise in this population and provide guidance to maximise impact and adherence longevity. Studies examining the use of probiotics and specific diets were limited, making conclusions and recommendation difficult, however, given research indicating the role of probiotics in neuroinflammation [54] and neuropathophysiological changes observed in Alzheimer's disease in those on a ketogenic diet [55], these research avenues should be further explored.

Beyond the interventions currently considered in the literature, improving sleep quality may be an important target to prevent cognitive decline, given the association with poor cognitive function in people living with HIV [56]. This is especially important given that sleep disturbance is commonly seen in both people living with HIV and older adults [7]. Interestingly, Vance et al. [57] conducted a focus group study looking at Cognitive Prescriptions, where lifestyle factors are addressed by daily goals in the form of a prescription for people living with HIV. This study showed a high acceptability and willingness to engage with such an intervention, suggesting a further avenue of future research.

It is of interest, that for most interventions included in this review, the main outcome was improvement in neuropsychological test scores. This is somewhat in contrast to recent community-based approaches that have documented a shift in patient priorities and made calls to develop local and feasible interventions which largely focus on improvement of HRQL and daily life for those with HIV and cognitive difficulties [17]. Consideration of the outcomes important to patients is an essential consideration for those designing interventions. Studies should incorporate designs which ensure patient perspectives and priorities are embedded throughout the research process. Improvements in neuropsychological test outcomes which do not translate to a meaningful improvement in patients' lives can be considered redundant from a person-centred perspective if they fail to address or improve the practical, everyday issues that matter most to patients, emphasising the need for a holistic approach that prioritises QoL and functional wellbeing. Developing interventions which assist individuals with activities of daily living and encourage social inclusion are important to facilitate independence and overall wellbeing. Indeed, in an era where people living with HIV are living longer it is important to also approach person-centred care from the perspective of maintaining or improving HRQL. Importantly, these considerations align with growing calls to include QoL as a 'fourth 90' in the updated UNAIDS 90-90-90 Testing and Treatment targets [58, 59]. A key question regarding future work in this area relates to whether studies should focus on those with cognitive symptoms, or those with measurable impairment on neuropsychological testing, who may not necessarily have manifested clinical symptoms as of yet. Given the complexity of neuropsychological testing, and the heterogeneity of cognitive batteries currently. used across studies, this may make application to the clinical setting more challenging. Despite this, clinical manifestation may be a late presentation of CI, and given the lack of consensus over how to quantify CI in the HIV population, using precise neuropsychological testing to detect CI may be

preferable to ensure studies capture all the possible participants affected. Establishing a standardised cognitive battery for use in experimental studies investigating CI in people living with HIV could standardise research to facilitate comparison between studies and help identify which interventions are the most valuable. A standardised battery could also be implemented into clinical practice for detecting those with any CI.

The scoping design of this review enabled the inclusion of diverse sources of evidence and provide an overview of the current evidence describing the non-pharmaceutical management of CI in people living with HIV. This review followed the PRISMA Sc-R [23] and the JBI Manual for Evidence Synthesis [60], ensuring gold standards for the design of a scoping review were met. Whilst there is certainly value in mapping the current literature, a limitation of this review was the heterogeneity of study designs, outcome measures and cognitive batteries used, making comparison within intervention type difficult. Additionally, although we aimed to include studies where objective CI detected at baseline was not primarily caused by co-morbidities, some studies identified mental health disorders such as depression amongst some participants. Whilst these were reported as secondary illnesses and not the cause of CI, the presence of these co-morbidities may have impacted the intervention outcomes. Most participants across studies were male, leading to an underrepresentation of female participants in this cohort. Many studies had small sample sizes, which limits the reliability of findings; further work should investigate the effects of promising interventions in larger populations.

Future studies should ensure implementation outcomes are assessed to ensure potential interventions are feasible, acceptable and appropriate. Cost effectiveness analyses may also be beneficial to determine which interventions may be suitable for widespread implementation. In conclusion, non-pharmaceutical interventions may represent a promising area for improving CI in people living with HIV. Work must be done to ensure interventions are effective and specific for people living with HIV, and crucially, embedded into HIV care settings to facilitate uptake.

## Supporting information

**S1 Appendix. Preferred Reporting Items for Systematic reviews and Meta-Analyses extension for Scoping Reviews (PRISMA-ScR) checklist.**
(DOCX)

**S1 Table. Demographic data from studies included in this review, used to calculate mean and standard deviation for age, and interquartile range for intervention length.**
(DOCX)

## Author Contributions

**Conceptualization:** Lucinda Stuart, Kate Alford, Jamie H. Vera.

**Formal analysis:** Lucinda Stuart, Kate Alford, Jamie H. Vera.

**Investigation:** Lucinda Stuart, Kate Alford, Jamie H. Vera.

**Writing – original draft:** Lucinda Stuart, Jamie H. Vera.

**Writing – review & editing:** Lucinda Stuart, Kate Alford, Jamie H. Vera.

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
