## [Decision Letter · Decision Letter 0]

14 Oct 2024

PONE-D-24-37701Non-pharmaceutical interventions for people living with HIV with cognitive impairment: a scoping reviewPLOS ONE

Dear Dr. Stuart,

Thank you for submitting your manuscript to PLOS ONE. After careful consideration, we feel that it has merit but does not fully meet PLOS ONE’s publication criteria as it currently stands. Therefore, we invite you to submit a revised version of the manuscript that addresses the points raised during the review process.

Please address the minor comments requested by the reviewers. Your review covers an important topic for people living with HIV and their brain health. This up to date review will be very informative for both the HIV healthcare professionals and the HIV community.

We look forward to receiving your revised manuscript.

Kind regards,

Lucette A Cysique, PhD

Academic Editor

PLOS ONE

Journal Requirements:

3. Thank you for stating the following in the Competing Interests section: “LS has no conflicts of interest. KA has no conflicts of interest. JHV has received honoraria and research grants from and been a consultant or investigator in trials sponsored by Merck, Janssen Cilag, Piramal and Gilead Sciences. He has received sponsorship to attend scientific conferences from Janssen Cilag, Gilead Sciences and AbbVie.”

We note that you received funding from a commercial source: Merck, Janssen Cilag, Piramal, Gilead Sciences and bVie.

Please provide an amended Competing Interests Statement that explicitly states this commercial funder, along with any other relevant declarations relating to employment, consultancy, patents, products in development, marketed products, etc. Within this Competing Interests Statement, please confirm that this does not alter your adherence to all PLOS ONE policies on sharing data and materials by including the following statement: "This does not alter our adherence to PLOS ONE policies on sharing data and materials.” (as detailed online in our guide for authors http://journals.plos.org/plosone/s/competing-interests). If there are restrictions on sharing of data and/or materials, please state these. Please note that we cannot proceed with consideration of your article until this information has been declared. Please include your amended Competing Interests Statement within your cover letter. We will change the online submission form on your behalf.

4. We note that your Data Availability Statement is currently as follows: “All relevant data are within the manuscript and in Supporting Information files.”

Please confirm at this time whether or not your submission contains all raw data required to replicate the results of your study. Authors must share the “minimal data set” for their submission. PLOS defines the minimal data set to consist of the data required to replicate all study findings reported in the article, as well as related metadata and methods (https://journals.plos.org/plosone/s/data-availability#loc-minimal-data-set-definition). For example, authors should submit the following data: - The values behind the means, standard deviations and other measures reported; - The values used to build graphs; - The points extracted from images for analysis. Authors do not need to submit their entire data set if only a portion of the data was used in the reported study. If your submission does not contain these data, please either upload them as Supporting Information files or deposit them to a stable, public repository and provide us with the relevant URLs, DOIs, or accession numbers. For a list of recommended repositories, please see https://journals.plos.org/plosone/s/recommended-repositories. If there are ethical or legal restrictions on sharing a de-identified data set, please explain them in detail (e.g., data contain potentially sensitive information, data are owned by a third-party organization, etc.) and who has imposed them (e.g., an ethics committee). Please also provide contact information for a data access committee, ethics committee, or other institutional body to which data requests may be sent. If data are owned by a third party, please indicate how others may request data access.

Additional Editor Comments:

Thank you for addressing the minor changes requested by the reviewers.

Reviewers' comments:

Reviewer's Responses to Questions

**Comments to the Author**

1. Is the manuscript technically sound, and do the data support the conclusions?

Reviewer #1: Yes

Reviewer #2: Yes

2. Has the statistical analysis been performed appropriately and rigorously? 

Reviewer #1: N/A

Reviewer #2: N/A

3. Have the authors made all data underlying the findings in their manuscript fully available?

Reviewer #1: Yes

Reviewer #2: Yes

4. Is the manuscript presented in an intelligible fashion and written in standard English?

Reviewer #1: Yes

Reviewer #2: Yes

5. Review Comments to the Author

Reviewer #1: Non-Pharmaceutical Interventions for People Living with HIV with Cognitive Impairment: A Scoping Review

I appreciate the opportunity to review this very interesting article. Overall, I found the article well written and to the point. Being familiar with this area, I agree with the conclusions. I only have a few suggestions/comments.

• Page 6, line 123 – remove comma after Despite.

• Page 7, lines 138 – Is this acceptability of interventions?

• Page 9, lines 188 – When starting a sentence with a number, it should be spelled out.

• Table with Byun intervention details – It was speed of processing and attention as standard, that is if there were deficits in those domains, otherwise other cognitive domains with corresponding training was conducted. That is the way for the TOPS study as well.

• Page 23, line 286 – remove comma after sessions.

• Page 23 – There are some HIV transcranial studies missing from this search.

o Fazeli, P. L., Woods, A. J., Pope, C. N., Vance, D. E., & Ball, K. K. (2019).

The effect of transcranial direct current stimulation combined with cognitive training on cognitive functioning in older adults with HIV: A pilot study. Applied Neuropsychology: Adult, 26(1), 36-47.

o Pope, C. N., Stavrinos, D., Vance, D. E., Woods, A. J., Bell, T. R., Ball, K. K., & Fazeli, P. L. (2018). A pilot investigation on the effects of combination transcranial direct current stimulation and speed of processing cognitive remediation therapy on simulated driving behavior in older adults with HIV. Transportation Research Part F: Psychology and Behavior, 58, 1061-1073.

o Cody, S. L., Fazeli, P. L., Crowe, M. G., Kempf, M-C., Moneyham, L., Stavrinos, D., & Vance, D. E. (2020). The effects of speed of processing training and transcranial direct current stimulation on global sleep quality and speed of processing in older adults with and without HIV: A pilot study. Applied Neuropsychology: Adult, 27(3), 267-278.

Reviewer #2: Non-pharmaceutical interventions for people living with HIV with cognitive impairment: a scoping review

This scoping review describes the types of and evidence relating to the non-pharmaceutical interventions for people living with HIV who are experiencing cognitive decline despite effective HIV treatment. In the context where CI is common and there is no specific pharmacological treatment, the topic is important. The choice of a scoping review as opposed to a systematic review is well justified and reporting guidelines are followed.

A few comments:

• ‘This indicates a role for non-pharmaceutical interventions, which aside from potentially improving neuropsychological outcomes may be effective at improving self-reported functional abilities and QoL’. And ‘It is of interest, that for most interventions included in this review, the main outcome was improvement in neuropsychological test scores. Recent community-based approaches have documented a shift in patient priorities and calls to develop local and feasible interventions which improve HRQL and daily life for those with HIV and cognitive difficulties’. Could the authors comment on this discrepancy and how the patient’s perspective and priorities could be better integrated in the design of non-pharmacological studies. Should studies focus on people with impairment on neuropsychological testing (and how would this eventually be transferred to the clinic setting as such testing is usually not available outside the research setting?), or on people with cognitive complaints? And what should the outcomes be? Currently, there is a lack of clarity about what the explanatory framework and how it influences study design. Is the hypothesis that improving performance on a cognitive test will improve function and quality of life, as is explicitly stated in the referenced paper regarding the ACTIVE study? Comment on this issue would enrich the paper.

• ‘Establishing a standardised cognitive battery for use in experimental studies investigating CI in people living with HIV would be extremely beneficial to future work, to facilitate comparison between studies and help identify which interventions are the most valuable’: this statement seems in contradiction with the point regarding patient’s’ priorities. Please elaborate.

• When reporting studies of cognitive training, it would be important to specify if the intervention was delivered in person or performed independently, and the documented adherence, especially if administered at home. While unsupervised administration is less costly, adherence to the intervention is an important (and problematic) aspect.

A few minor points:

• Approximately 14% of people living with HIV experience objective CI: the study quoted was conducted among adults >50 years, a group more at risk: please mention the age group

• ‘The Training on Purpose Study (TOPS) examined 20 hours of individualized CCT over 262 12 weeks, targeting specific cognitive deficits identified at baseline. While the study did not reverse a HAND diagnosis for most participants, it did improve global clinical 264 ratings’. (ref missing)

• ‘In contrast to the findings of TOPS, this study found lower global clinical rating scale scores in the control group both immediately and one year post-intervention than in both arms’: This statement is not clear. Lower than what? The original paper states: Contrary to expectations, those in the control group had slight improvements in global function.

• The iSTEP intervention: there is no mention of the findings on cognition. Please add

and a couple of typos:

• Despite, some HIV guidelines: no comma

• simulate further consideration: stimulate

6. PLOS authors have the option to publish the peer review history of their article (what does this mean?). If published, this will include your full peer review and any attached files.

Reviewer #1: No

Reviewer #2: **Yes: **Marie-Josée Brouillette

---

## [Author Response · Author response to Decision Letter 0]

1 Nov 2024

Addressed in the "Response to Reviewers" file in this submission.

---

## [Editor Report · Decision Letter 1]

7 Nov 2024

Non-pharmaceutical interventions for people living with HIV with cognitive impairment: a scoping review

PONE-D-24-37701R1

Dear Dr. Stuart,

We’re pleased to inform you that your manuscript has been judged scientifically suitable for publication and will be formally accepted for publication once it meets all outstanding technical requirements.

Kind regards,

Lucette A Cysique, PhD

Academic Editor

PLOS ONE

Additional Editor Comments (optional):

Thank you for addressing the reviewers' comments.

This paper represents an important contribution to the field of NeuroHIV as well as ageing and HIV.

---

## [Editor Report · Acceptance letter]

14 Nov 2024

PONE-D-24-37701R1 

PLOS ONE

Dear Dr. Stuart, 

I'm pleased to inform you that your manuscript has been deemed suitable for publication in PLOS ONE. Congratulations! Your manuscript is now being handed over to our production team.

Kind regards, 

on behalf of

Dr. Lucette A Cysique 

Academic Editor

PLOS ONE